# Multiomic characterization of disease progression in mice lacking dystrophin

**Mirko Signorelli**[1]*, **Roula Tsonaka**[2], **Annemieke Aartsma-Rus**[3], **Pietro Spitali**[3]*

**1** Mathematical Institute, Leiden University, Leiden, The Netherlands, **2** Department of Biomedical Data Sciences, Leiden University Medical Center, Leiden, The Netherlands, **3** Department of Human Genetics, Leiden University Medical Center, Leiden, The Netherlands

* msignorelli.papers@gmail.com (MS); p.spitali@lumc.nl (PS)

## Abstract

Duchenne muscular dystrophy (DMD) is caused by genetic mutations leading to lack of dystrophin in skeletal muscle. A better understanding of how objective biomarkers for DMD vary across subjects and over time is needed to model disease progression and response to therapy more effectively, both in pre-clinical and clinical research. We present an in-depth characterization of disease progression in 3 murine models of DMD by multiomic analysis of longitudinal trajectories between 6 and 30 weeks of age. Integration of RNA-seq, mass spectrometry-based metabolomic and lipidomic data obtained in muscle and blood samples by Multi-Omics Factor Analysis (MOFA) led to the identification of 8 latent factors that explained 78.8% of the variance in the multiomic dataset. Latent factors could discriminate dystrophic and healthy mice, as well as different time-points. MOFA enabled to connect the gene expression signature in dystrophic muscles, characterized by pro-fibrotic and energy metabolism alterations, to inflammation and lipid signatures in blood. Our results show that omic observations in blood can be directly related to skeletal muscle pathology in dystrophic muscle.

## Introduction

Duchenne muscular dystrophy (DMD) is an X-linked, recessively inherited neuromuscular disorder caused by lack of dystrophin, the protein product of the *DMD* gene[1]. Multi-exon deletions and duplications[2,3] are the most frequent mutation type, leading to a shift in the reading frame and the presence of premature termination codons. Although rare, DMD affects thousands of children world-wide[4–6]. Boys with DMD show delayed motor development, and lose the ability to walk independently on average before the age of 12 [7]. Despite large investments in the development of therapeutic compounds ranging from small drugs [8] up to complex biologics [9], multiple drugs failed to show clinical benefit in clinical trials [10], and only a few treatments made it to the market (with disputed efficacy) [11].

The lack of efficacy observed in unsuccessful clinical trials is due to factors such as low drug potency and large variability in disease trajectories across patients, resulting in underpowered clinical trials [10]. The short duration of clinical trials and the lack of surrogate endpoints able to anticipate future clinical benefit further reduced the chance of trials being successful. However, not all failures are due to low drug potency and poor clinical trial designs: multiple

**Data Availability Statement:** The data analyzed in this study are available from the following repositories: 1. RNA-seq data (both blood and muscle): Gene Expression Omnibus repository,

accession id GSE132741. 2. Lipids: S3 File. 3. Metabolites: Supplementary Material, S3 and S4 Tables of Tsonaka et al. (2020), URL: https://doi.org/10.1093/hmg/ddz309.

**Funding:** PS: Duchenne Parent Project NL. Website: https://duchenne.nl/. Project number 16.006. The funders had no role in study design, data collection and analysis, decision to publish, or preparation of the manuscript.

**Competing interests:** The authors have declared that no competing interests exist.

failures can be traced back to the preclinical studies, where proof of concept studies have been used to justify a study in human without clear efficacy in preclinical studies. It is therefore important to be able to identify biomarkers that can be used to monitor disease progression and drug efficacy in both preclinical and clinical settings [12–16]. Multiple groups have therefore embarked in the identification of biomarkers that associate with disease progression, mostly focusing on lowly- or non-invasive sample matrices such as serum and urine [17–21]. Biomarkers associated with increased odds of loss of ambulation have been identified in patients, alongside with safety and efficacy biomarker related to treatment with corticosteroids [22,23]. Analysis of proteins, miRNAs and metabolites has been attempted to identify biomarkers able to discriminate between *mdx* mice and WT animals [21,24–27] and between DMD patients and healthy age-matched controls [28–30]. More recently, longitudinal proteomic studies have led to the identification of biomarkers associated with exposure to drugs prescribed as standard of care such as corticosteroids, as well as with patients performance over time [22,31,32].

However, preclinical studies published so far show a number of limitations. First, they focused on a single omic direction (e.g., transcriptomics, proteomic or metabolomic), providing a limited understanding of how transcriptional changes relate to the downstream omic observations such as protein and metabolites, which are more directly linked to phenotypic observations. Second, the vast majority of studies focused on cross-sectional comparisons, without providing an overall picture of how the gene expression, metabolites and lipids signatures evolve over time. Finally, biomarker studies focused mostly on single sample matrices such as plasma or serum or urine or muscle, omitting to explore correlation across omic features between dystrophic muscles and more accessible biofluids.

To overcome these limitations, we designed a study where different omic readouts were captured longitudinally and across tissues. This design aimed to achieve three goals: first, to evaluate similarity of contemporary expression patterns across different omic layers (RNA, metabolome and lipidome) and tissues (blood and muscle); second, to evaluate whether changes over time are consistent across different omic layers (RNA, metabolome and lipidome) in blood; third, to understand whether observations in muscle may be connected to observations in blood. The study design included 3 different dystrophic mouse models and healthy mice. We collected up to 5 measurements, spaced over 7 months, for each mouse, significantly extending over the period typically taken as reference for pre-clinical development of therapeutic compounds [33]. The study yielded 4 different omic datasets: RNA-seq of muscle samples (1 sample per mouse), RNA-seq of blood samples, metabolomics of plasma samples, and lipidomics of plasma samples (up to 5 samples per mouse).

A separate characterization of each of these omic datasets has been presented in previous publications [34–39]. Muscle RNA-seq showed that a considerable number of genes were differentially expressed between WT mice and the three dystrophic groups [34,36]. We showed how these genes related to the pathophysiological changes occurring in dystrophic muscle, such as impaired energy metabolism, inflammation, fibrosis and tissue remodeling. Longitudinal analysis of blood RNA-seq further revealed that differences between mouse groups were stronger at week 6, where some overlap was observed with the pathways known to be affected in muscle, and progressively decreased at later weeks [34]. The analysis of lipids in plasma showed clear alterations in glycerolipids and glycerophospholipids and more specifically triglycerides, perhaps connected to the fiber type shift observed in muscular dystrophies [38]. Finally, analysis of metabolites in plasma showed how energy metabolism (such as glutamine, creatine) and muscle buffering (carnosine) are affected in dystrophic mice [35].

The results from those single-omic analyses naturally raise questions on which biological observations are represented across multiple omics, and whether it is possible to identify commons trends accounting for differences between mouse groups across the different omics and

different disease phases. To address these questions, we applied Multi-Omics Factor Analysis (MOFA) to the longitudinal multiomic dataset collected [40]. MOFA is a statistical method that can be used to summarize large multiomic datasets with a small number of latent factors (LFs) that retain as much of the variance of the original data as possible. Specifically, MOFA estimates a set of LFs that are shared across multiple omic datasets, making it possible to identify sources of variation that are common to some or all of the omic sources. By applying MOFA to our data, we were able to show the presence of common expression patterns and trends both across different omics and between blood and muscle.

## Materials and methods

### Description of the experiment

Our experiment (Fig 1A) involved 40 mice belonging to 4 groups: wild type (WT), *mdx*, *mdx+ +* and *mdx+-*. *Mdx* mice shared the same genetic background of WT mice (BL10), whereas mice in the remaining groups carried either 1 (*mdx+-*) or 2 (*mdx++*) functional copies of the utrophin paralog gene on a mixed genetic background. All mice were male and entered the experiment at 4 weeks of age. Mice were kept in individually ventilated cages, fed ad libitum with chow, and had free access to water. Blood samples were obtained via the tail vein at weeks 6, 12, 18 and 24, and from the eye at week 30; mice were fasted for 4 to 6 hours before sampling. To alleviate discomfort after blood collection at 6, 12, 18 and 24 weeks, the tails were treated with a lidocaine solution. For the 30 weeks blood collection, mice were anesthetized using isoflurane before blood collection. Mice were then sacrificed by cervical dislocation after the last blood sample was collected at 30 weeks of age, with the exception of a few mice that had to be sacrificed before the end of the experiment in line with pre-specified humane endpoints. The tibialis anterior muscles were harvested and further analyzed by H&E staining and RNA-sequencing. H&E staining was performed as previously described [41]. Representative pictures of the H&E stained sections are presented in S1 Fig. The experiment was approved by the local animal welfare committee (DierExperimentenCommissie Academisch Ziekenhuis Leiden, protocol number 13154).

### Generation of RNA-sequencing data, filtering and normalization

RNA was purified from the tibialis anterior muscle and from full blood (not cell free RNA) obtained in RNeasy Protect Animal Blood Kit (Qiagen, Cat. N. 73224). For blood RNA, a globin depletion step was performed using GLOBINclear Kit for mouse/rat (Thermo Fisher, Cat. N. AM1981). Sample preparation was performed using the TruSeq Poly-A v2 Kit (Illumina, San Diego). The BIOPET Gentrap in-house pipeline was used to analyze the sequencing data. Quality control was performed using FastQC and MultiQC. Data were aligned to the mouse reference genome GRCm38 using STAR aligner version 2.3.0e with an average of 81.6% alignment ratio as previously described [34]. Before normalization, we filtered out lowly expressed genes in each of the two datasets by retaining only genes with at least five counts per million (cpms) in at least 10% of the samples. The data were subsequently normalized using the Trimmed Mean of M values method [42] and converted to normalized cpms.

### Generation and normalization of metabolite and lipid data

Analysis of metabolites and lipids was performed by liquid chromatography combined with mass spectrometry (LC-MS). The preparation and acquisition of the metabolite and lipids levels were previously described [35,38]. In short, plasma samples were introduced into a Transcend 1250 LC (Thermo Fisher Scientific) fitted with a Sequant ZICpHILIC 5 μm, 2.1 × 150

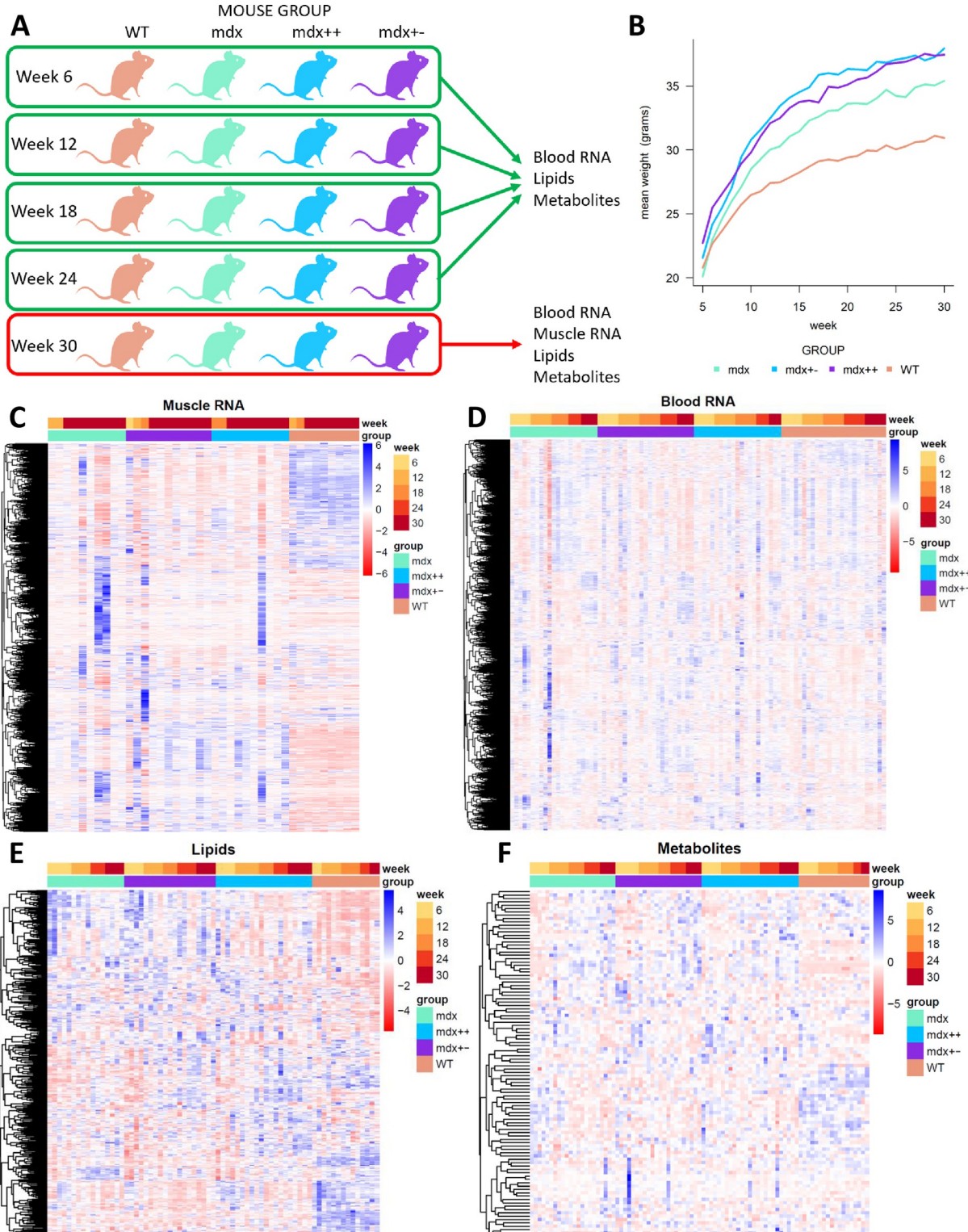

**Fig 1. Overview of the multiomic data analysed in this article.** A) Schematic representation of the experiment. WT, *mdx*, *mdx++ and* mdx +- mice were monitored for 30 weeks. Blood samples were drawn every 6 weeks, allowing measurement of RNA, lipids and metabolites. At week 30, mice were sacrificed and muscle samples drawn, allowing measurement of RNA expression in muscle. B) Weekly mean weight (in grams) of the mouse groups, measured from week 5 to week 30. C-F) Heatmaps showing the distribution of blood RNA (panel C), muscle RNA (panel D), lipids (panel E) and metabolites (panel F) across samples.

mm column (Merck) for the metabolite data, and a kinetex C8 2.6 μm 2.1 × 150 mm column (Phenomenex) for the lipid data. This was then coupled to a Q-Exactive mass spectrometer (Thermo Fisher Scientific) in both positive and negative ionization modes separately. Data analysis (peak picking and features annotation) was performed using TraceFinder 3.1 (Thermo Fisher Scientific) for the metabolites, and XCMS for the lipids. Both the metabolite and the lipid data were normalized using the Probabilistic Quotient Normalization [43].

## Multi-Omics Factor Analysis (MOFA)

Integration of the four omic datasets was performed using Multi-Omics Factor Analysis (MOFA) [40]. In short, MOFA infers common latent factors (LFs) that are shared across different omic datasets, called views. More specifically, MOFA decomposes the data matrix $Y_m$ for a given view $m$ as follows:

$$Y_m = ZW_m + E_m,$$

where Z contains the LFs that are shared across all views, $W_m$ is a matrix of factor loadings that are specific to each view and reflect the weight with which each of the molecules in view $m$ contribute to each LF, and $E_m$ is a matrix of error terms (S2 Fig).

To remove differences of scale between RNA-seq and mass spectrometry data, all genes, metabolites and lipids in the different omic views were scaled to unit variance.

In theory, MOFA is able to handle missing data. However, the impossibility to measure both blood RNA-seq and lipids and metabolites in samples from the same mouse led to a high proportion of missing data; with such a high percentage of missing data points and lack of overlap in samples between blood RNA-seq on the one hand and lipids and metabolites on the other hand, it was impossible to achieve convergence of MOFA's estimation algorithm. To address this computational problem, we proceeded to match blood RNA-seq samples to metabolite and lipid samples using the following matching variables: mouse identifier, mouse group and week of sample collection. When this matching led to the availability of two muscle RNA-samples collected at the same time point, we averaged the normalized cpms of genes in muscle over the two samples.

## Model training and selection

MOFA's estimation algorithm requires the specification of an initial number of components, of a random seed, and of a threshold on the percentage of variance explained by each component that is used to drop components from the model. Different choices of these starting values typically affect the number of factors at which the estimation algorithm converges. Additionally, the non-convexity of the parameter space of MOFA models implies that model fits obtained fits with the same number of components need to be compared to determine which one has the highest Evidence Lower Bound (ELBO). In practice, these two issues raise the problem of implementing a strategy for model selection.

To address this methodological issue, we considered eight different variance thresholds (from 2% to 20%), and for each threshold we computed 100 model fits, each obtained from a different random seed. This computation yielded 800 alternative model fits, with a number of factors that ranged from 1 to 18. To perform model selection, we first grouped the model fits by number of factors, and within each subgroup we selected the model with the highest ELBO. By doing so, we obtained 18 model fits, each of which is the best model fit for a given number of factors. Then, we compared these 18 models by drawing a line plot that shows the total percentage of variance explained by MOFA, and the percentages of variances explained in each view, as a function of the number of factors (Fig 2B and 2C). In order to balance the complexity and explanatory power of the model, we selected the MOFA model with 8 factors, as we observed

not only that this model explained a large amount of the total variance (78.8%), but also that the total percentage of variance explained started to level off when more than 8 factors are considered (Fig 2B). The factor loadings of the selected model are provided in S2 File.

## Results

### Experimental setup

We designed a longitudinal experiment (Fig 1A) involving wild type (*WT*) mice and three groups of dystrophic mice (*mdx*, *mdx utrn++*, *mdx utrn+-*). Mice entered the experiment at 4

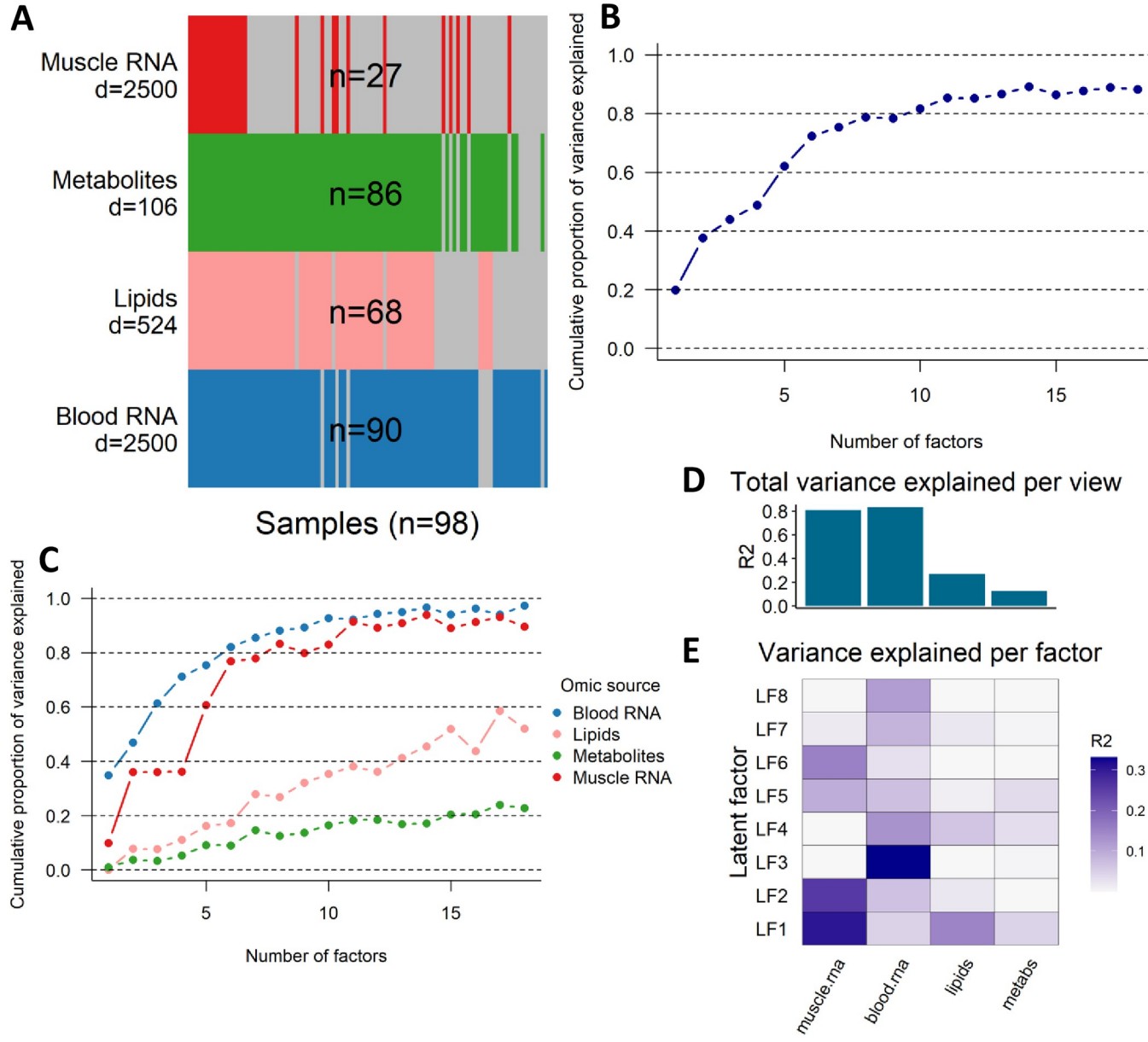

**Fig 2. Summary of the multi-omics factor analysis.** A) Data overview showing the number of samples (n) and molecules (d) available in each omic view. A grey bar indicates that the sample is missing in the given omic view. B) Cumulative proportion of variance explained for MOFA fits with increasing number of factors. C) Cumulative proportion of variance explained by omic view for MOFA fits with increasing number of factors. D) Total percentage of variance explained (R2) by omic view (top), and percentage of variance explained by each latent factor in the different omic views (bottom) for the selected MOFA model. E) Comparison of the distribution of the factor loadings across different omic views.

weeks of age. Starting from week 6, a blood sample was collected from each mouse every 6 weeks, up to week 30. With the exception of a few mice that had to be sacrificed before the end of the experiment in line with pre-specified humane endpoints, most mice were sacrificed at week 30; tibialis anterior muscle samples were collected from each sacrificed mouse. Dystrophic mice showed increased weight compared to healthy mice (Fig 1B), in line with previous reports [41,44,45].

We employed mass spectrometry to measure the abundance of lipids and metabolites in plasma samples, and RNA-seq to quantify gene expression in blood and muscle samples. A total of 271 omic data points across the 4 omic views were included in the study. Visual inspection of the data across genotypes and time points with heatmaps showed less variation across mice in the WT group compared to the other groups, and stronger dystrophic signatures for the muscle RNA-seq and lipidomics views, while somewhat less signal was visible for metabolites and blood RNA-seq (Fig 1C–1F).

## Multiomic integration through multiomic factor analysis

In order to integrate the different omics with MOFA, we constructed 98 multiomic profiles by matching samples in the 4 omic views by mouse identifier and week (Fig 2A). A total of 524 lipids, 106 metabolites, and 24450 genes were detected in the experiments. To ensure that the metabolomic and lipidomic views were not underrepresented when fitting MOFA, we reduced the number of genes in the two RNA-seq views, retaining only the 2500 genes that had larger variance in each view. This selection was based on a robustness check of different MOFA fits with 1000 genes, 2500 genes and with all of the expressed genes (i.e., all genes that were retained after the filtering step). This robustness check showed that the total variance explained by MOFA when all genes expressed in muscle and blood were included in the model was comparable to the one from models including only the top 2500 or top 1000 genes (S3 Fig). However, we observed that estimating MOFA with a larger number of genes in the blood and muscle RNA-seq views led to a significant decrease in the percentage of variance explained in the lipidomic and metabolomic views, without increasing the variance explained in the RNA-seq views (S4 Fig). To ensure that the inferred multiomic factors could summarize sufficient variance also for the omic views with less features, we therefore proceeded to interpret the model estimated using the top 2500 genes expressed in blood and muscle RNA (Fig 2C).

To train and select a MOFA model, we considered several starting values and stopping criteria, obtaining 800 model fits with a number of LFs that ranged from 1 to 18 (more details provided in the Model Selection section). For each given number of LFs, we selected the model with the largest ELBO. Models with different number of factors were compared by visualizing the total percentage of variance explained by each model (Fig 2B), and the percentage of variance explained in each omic view (Fig 2C). The total percentage of variance explained by the LFs was generally higher for the RNA-seq datasets, and lower for metabolites and lipids, irrespective of the number of components. This pattern is probably due to the higher variation typically observed in RNA-seq datasets, as well as to the lower number of features in the lipidomic and metabolomic views.

After a steep increase in the explanatory power up to 8 LFs, the total percentage of variance explained starts to level off (Fig 2B). In order to strike a balance between model complexity and explanatory power, we thus selected the best MOFA model (largest ELBO) with 8 LFs, which explains 78.8% of the total variance present in the multiomic dataset (see the Model selection section for more details). The selected model explained 83.3% of the variance in the muscle RNA-seq view, 88.1% in the blood RNA-seq view, 26.9% in the lipidomic view, and 12.5% in the metabolomic view (Table 1).

**Table 1. Percentage of variance explained by each LF in each omic view.**

| Latent Factor | Blood RNA | Lipids | Metabolites | Muscle RNA | Overall |
|---|---|---|---|---|---|
| 1 | 4.7% | 15.1% | 4.6% | 30.6% | **17.2%** |
| 2 | 6.9% | 2.1% | 0.0% | 25.7% | **14.7%** |
| 3 | 33.1% | 0.0% | 0.5% | 0.0% | **14.7%** |
| 4 | 13.1% | 6.4% | 3.3% | 0.0% | **6.5%** |
| 5 | 7.1% | 1.2% | 3.6% | 9.5% | **7.6%** |
| 6 | 2.9% | 0.0% | 0.0% | 15.5% | **8.2%** |
| 7 | 8.6% | 2.1% | 0.5% | 1.8% | **4.8%** |
| 8 | 11.6% | 0.0% | 0.0% | 0.2% | **5.2%** |
| **Total** | **88.1%** | **26.9%** | **12.5%** | **83.3%** | **78.8%** |

## Overview of the latent factors (LFs)

The percentage of variance explained in each omic view by the 8 LFs is presented in Table 1 and visualized in Fig 2D. LF1 and LF5 explain a non-negligible percentage of variance in each omic view, and they can thus be seen as factors that are common to all views; together, they explain 24.8% of the total variance. LF2, LF4 and LF7 are common to 3 omic views, whereas the remaining factors are more specific to one (LF3 and LF8) or two (LF6) views. The fact that most LFs explain variance across multiple omic views and are therefore not specific single omic views provides an indication that data are rather strongly correlated across views, and that it is possible to identify common patterns across the different omic views.

To understand how the LFs relate to the dystrophic phenotype and disease progression, we proceeded to assess the relationship between the factor loadings and mouse group and time (Fig 3A and 3B, Tables 2 and 3). LF1 was found to be strongly associated both with mouse group ($R^2$ = 0.335, adjusted p-value < 0.0001) and time ($R^2$ = 0.434, adjusted p-value < 0.0001). LF4 and LF5 were strongly associated with time ($R^2$ = 0.374 and adjusted p-value < 0.0001 for LF4; $R^2$ = 0.295 and adjusted p-value = 0.0129 for LF5). For the statistically significant differences (adjusted p < 0.05) in Table 3, we further tested differences in the expected value of the LFs between mouse groups and weeks (S2 File). As concerns LF1, differences were significant for most pairs of groups; the only exception to this was *mdx++* vs *mdx+-*, for which no significant difference was found. Moreover, weeks 6 and 30 were found to be significantly different from all other weeks, whereas no significant difference was found between weeks 12, 18 and 24. A similar pattern with respect to week differences was observed for LF5. Instead, for LF4 the significance of differences in mean between weeks increased as the distance between weeks increased, consistently with the increasing pattern visible in Fig 3B. When considered jointly, group and week explain 81.4% of the variability of LF1, 45.2% of the variability of LF4, and 44.5% of the variability of LF5 (Table 2). The increased variance explained by the model with the interaction compared to the models with only main effects is suggestive of the extent to which a latent factor may capture diverging trajectories between healthy and dystrophic mice, and points out the importance of jointly considering time and mouse group when interpreting the LFs. Because our interest lies both in the identification LFs that explain variability across several omic views, and in relating such factors to the different dystrophic phenotypes (group) and disease progression (time), hereafter we will focus our attention on LFs 1, 4 and 5.

## Latent factors 1, 4 and 5 are associated with disease progression and a dystrophic phenotype

LF1 separates rather well WT mice from *mdx++* and *mdx+-*, with *mdx* mice laying in between the other groups (Fig 3A). Thus, this factor captures both differences between healthy and

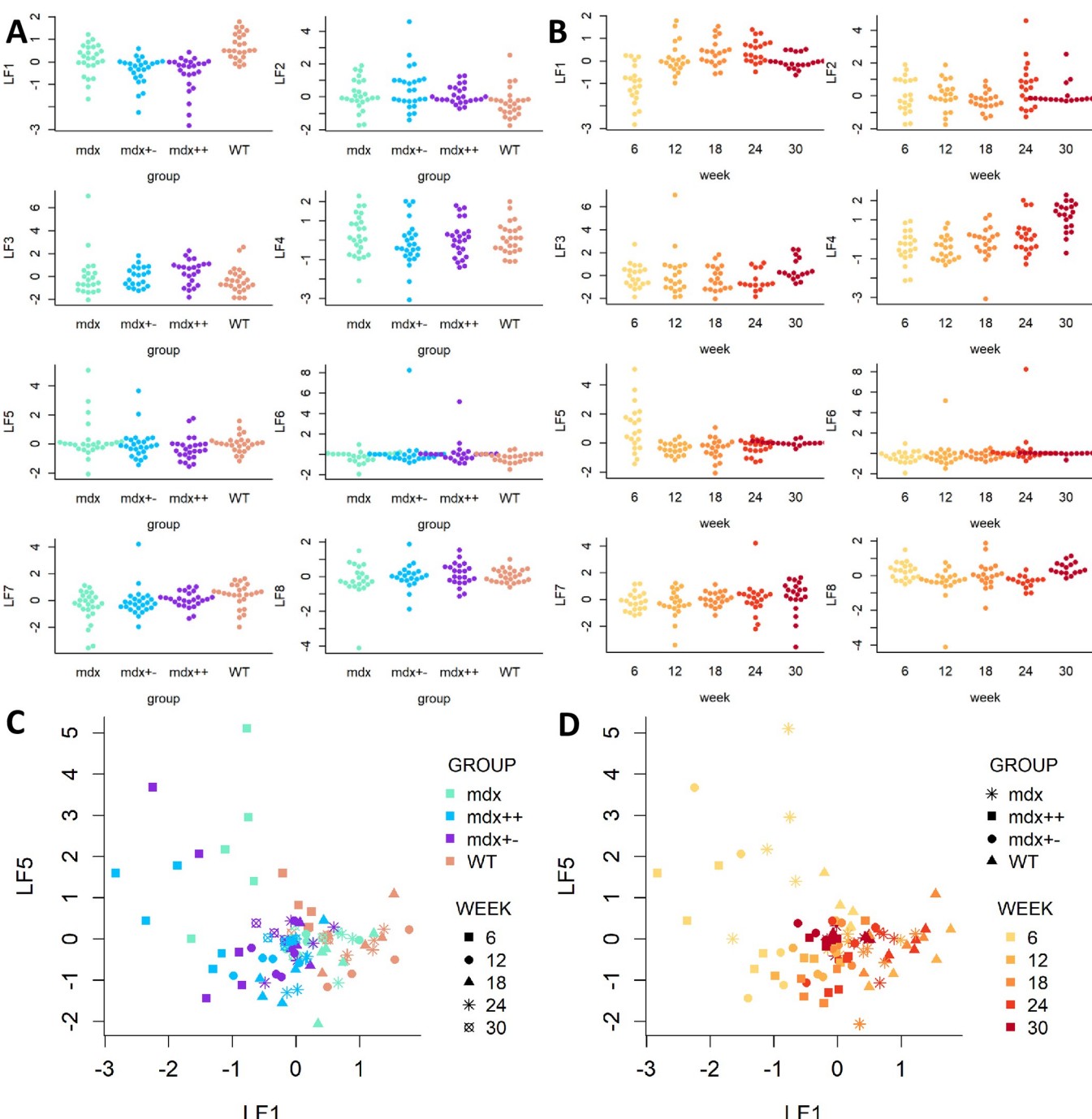

**Fig 3. Latent space representation of the 8 MOFA factors.** A) Beeswarm plots representing the distribution of each latent factor across the 4 mouse groups. B) Beeswarm plots representing the distribution of each latent factor by week. C) Latent space representations of factors 1 and 5. The colour of dots denotes mouse group, and the shape denotes the week of sampling. D) Latent space representations of factors 1 and 5. The colour of dots denotes week of sampling, and the shape denotes mouse group.

dystrophic mice (WT vs the 3 dystrophic groups) and differences in genetic background (WT and *mdx* vs *mdx++* and *mdx+-* mice). Moreover, LF1 also separates samples gathered at week 6 from samples obtained at later time points (Fig 3B); this separation is interesting, as week 6 typically represents the phase of intense skeletal muscle damage and repair, which is followed

**Table 2. Percentage of variance of each LF that is explained ($R^2$) by mouse group (column 2), week (column 3), and by both mouse group and week (based on a linear model comprising interaction terms, column 4).**

| Latent Factor | Variance explained by group | Variance explained by week | Variance explained by group*week |
|---|---|---|---|
| 1 | 33.5% | 43.4% | 81.4% |
| 2 | 9.0% | 8.7% | 36.0% |
| 3 | 4.2% | 6.2% | 28.8% |
| 4 | 2.7% | 37.4% | 45.2% |
| 5 | 4.3% | 29.5% | 44.5% |
| 6 | 3.2% | 5.4% | 17.4% |
| 7 | 9.2% | 3.3% | 22.7% |
| 8 | 2.8% | 19.3% | 36.2% |

by a stabilization of this; the fact that this separation is visible in our dataset despite the lack of muscle samples at week 6 indicates that biomarkers of this phase, which primarily involves muscle, can be observed in blood, and especially in the lipidomic view, which is the main blood contributor to LF1 (Table 1 and Figs 2E and 3D).

LF5 is the other LF that explains a relevant percentage of variance in each view. This factor is mostly linked to time, and mostly separates samples from dystrophic mice collected at week 6 from all the other data points (Table 2 and Fig 3A and 3B). Visualization of the multiomic samples in the space defined by LF1 and LF5 shows that when considered jointly, the two LFs can separate extremely well week 6 samples from dystrophic mice from the later samples from dystrophic mice, as well as from all samples from WT mice (Fig 3C and 3D). Finally, LF4 appears to be positively correlated with time across all groups, reflecting a natural growth effect between 6 and 30 weeks of age observed in all mice.

## Multiomic dystrophic signature (LF1)

LF1 is the factor that best separates dystrophic mice from healthy ones (Fig 3A). This LF explains a considerable percentage of variance in the muscle RNA-seq (30.6%) and lipid (15.1%) views, and a relevant but lower percentage in the blood RNA-seq (4.7%) and metabolomic (4.6%) views. Fig 4A shows the 20 molecules that more strongly contribute to LF1 in each omic view. The large contribution of muscle RNAseq and lipidomic views to LF1 is reflected in the heatmaps where a clear separation between dystrophic and healthy mice is visible in the muscle RNA-seq (Fig 4B) and in the lipidomic data at all time points (Fig 4D). In blood RNA-seq group differences are mainly visible at early time points such as weeks 6 and

**Table 3. P-values and adjusted p-values for the F test of no difference in the mean of each LF between mouse groups and week.**

| Latent Factor | Group differences | | Week differences | |
|---|---|---|---|---|
| | p-value | BH adjusted p-value | p-value | BH adjusted p-value |
| 1 | < 0.0001 | **< 0.0001** | < 0.0001 | **< 0.0001** |
| 2 | 0.0279 | 0.0743 | 0.4128 | 0.5504 |
| 3 | 0.2994 | 0.4791 | 0.5638 | 0.5638 |
| 4 | 0.4559 | 0.4856 | 0.0000 | **< 0.0001** |
| 5 | 0.2333 | 0.4667 | 0.0048 | **0.0129** |
| 6 | 0.3723 | 0.4856 | 0.1294 | 0.2070 |
| 7 | 0.0251 | 0.0743 | 0.1215 | 0.2070 |
| 8 | 0.4856 | 0.4856 | 0.5541 | 0.5638 |

12, where gene expression levels in dystrophic mice are higher compared to WT mice and to later time points for almost all top 20 genes (Fig 4C). Differences between WT and dystrophic mice can also be observed in the metabolomic view, even though the separation is less evident in this view (Fig 4E).

To understand the information provided by LF1, we looked into the function of genes in the muscle RNA-seq view and lipids that are more strongly associated with LF1. Genes contributing to LF1 suggest that this factor captures the dystrophic phenotype. Multiple genes with high loading in LF1 relate to muscular dystrophy with the *Dmd* gene ranking in top 100 genes, and all 3 Col6a genes (Col6a1, Col6a2 and Col6a3 genes) responsible for Ulrich muscular dystrophy are present in the top 40 genes in LF1 (S5 Fig). The association of LF1 with dystrophic muscle is further supported by the presence of genes that have been shown to be histological biomarkers of muscle fibrosis, regeneration and inflammation/immune response. Example of known fibrosis markers present in LF1 are *Cola1*, *Cola2*, *Col3a1*, *Bgn*, *Fn1*, *Spp1* and *Ftl1* genes [46–51]. Genes involved in muscle regeneration (*Myl4*, *Mybph*, *Sparc*, *Gpx1*, *Myod1 and Myog*) and immune response (*Lyz2*, *Spp1*, *Ctsb*) were also represented in LF1 [52]. We further looked into whether the 883 genes whose loadings have absolute value above 2 had previously been associated reported to be associated with neuromuscular disorder. After converting the mouse gene IDs to human gene IDs with Biomart, we obtained a list of 840 genes. 82 of these 840 genes are known to cause neuromuscular conditions (S3 File). For the remaining 758 genes, we checked whether they have been previously associated with muscular phenotypes through Euretos; 257 genes have indeed previously reported to be associated with muscular phenotypes, whereas the remaining 507 genes have not been associated with a muscular phenotype so far (S4 File).

Pathway analysis confirmed the association of LF1 with dystrophic features with enrichment for fibrotic (such as collagen fibril organization and extracellular matrix organization) and mitochondrial processes (mitochondrial respiratory chain complex I assembly and respiratory electron transport chain) together with pathways involving lipid metabolism (fatty acid metabolic process, fatty acid beta-oxidation and lipid metabolic process; S6A and S7 Figs). To understand whether the same genes in blood and muscle contribute to LF1, we compared the loadings of LF1 for the genes that are shared by these 2 views. The correlation between loadings in the two views is extremely low, and directional changes are consistent only for part of the genes such as *Igfbp4*, *Col1a2* and *Sparc* (S8 Fig), supporting the use of *Col1a2* and *Sparc* blood gene expression as a proxy for muscle expression for the same genes. Pathway analysis for the blood genes showed mostly the inflammatory component of the disease (S6B and S7 Figs).

The contribution of lipids metabolism evident in the gene expression signature is consistent with the data in the lipidomic view, where glycerolipids (triacylglycerols and diacylglycerols) and glycerophospholipids are the most abundant species among the lipids with high loadings in this factor. Some sphingomyelins were also shown to be reduced in dystrophic mice plasma compared to healthy controls in line with the reduced expression levels of sphingomyelin synthase (*Sgms1*) in muscle (also loading high in LF1). Among the metabolites, the nicotinamide N-oxide (which relates to oxidation/reduction reactions, cell proliferation and fatty acid oxidation) showed the highest loading. Pathway analysis of metabolites highlighted processes connected to protein translation with pathways such as tRNA charging and aminoacyl-tRNA biosynthesis, which could be connected to the effect of nicotinamide N-oxide reduction on cell proliferation (S6C Fig). Despite being in the top 20 metabolites and a precursor of nicotinamide N-oxide synthesis, tryptophan levels did not seem to be related to the 6 weeks time point.

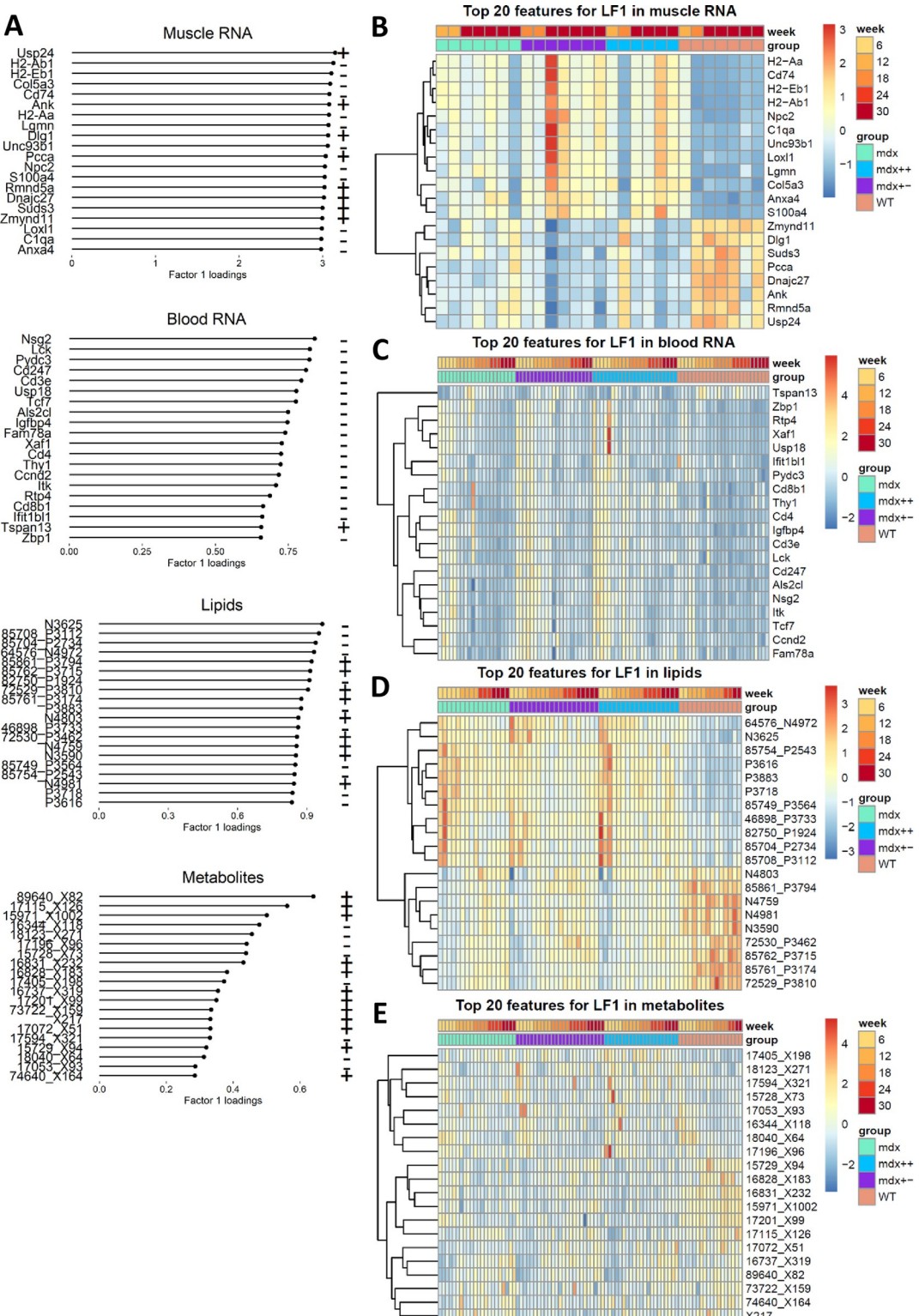

**Fig 4. MOFA factor 1: Top 10 molecules by factor weight in each omic view.** A) Factor 1 weights for the top 10 genes, lipids and metabolites by factor weight in muscle RNA, blood RNA, lipids and metabolites. B-E) Heatmaps showing the distribution of the top 10 molecules by factor weight in muscle RNA (panel B), blood RNA (panel C), lipids (panel D) and metabolites (panel E) across samples.

## Dynamic changes (LF5 and LF4)

Besides LF1, also LF4 and LF5 are associated with changes observed over time (Table 2 and Fig 3B). Specifically, the distribution of LF5 mostly differs at week 6, while LF4 exhibits an increasing trend over time (Fig 3B). The two major contributors for LF5 are muscle RNA-seq (9.5% of variance explained by LF5) and blood RNA-seq (7.1%), followed by metabolites (3.6%) and lipids (1.2%). Regulation of muscle contraction was the only significant pathway in the muscle RNA-seq (S7 Fig). Although muscle is the strongest contributor to LF5, the fact that most muscle RNA-seq samples were obtained at week 30 makes it apparent that differences between week 6 and the later weeks cannot be due to muscle samples; it is however possible that the latent factor describes part of the dystrophic signature that is shared across blood and muscle and that we capture in muscle only at 30 weeks because of our sampling strategy. Heatmaps showed that the separation of the 6 weeks time points is mostly due to blood RNA-seq data (Fig 5B–5E). The difference between week 6 and later time points is visible only in dystrophic groups and absent in *WT* mice, so it is likely that genes contributing to this factor relate to the disease rather than to a growth or early adulthood. Pathway analysis of blood RNA-seq highlighted pathways previously connected to muscular dystrophies such as autophagy of mitochondrion (S7 Fig). Heatmaps for the top 20 lipids and metabolites (Fig 5D and 5E) showed that these views contribute more to the separation of the groups rather than to the 6 weeks time point separation. Pathway analysis did not highlight any significant pathway for the metabolomic view, however top features included metabolite connected to the muscle buffering capacity such as carnosine and metabolism of nucleotides. LF4 showed a linear relationship with time. All omic views except for muscle contribute to LF4, with blood RNA-seq being the single largest contributor. Interestingly, pathway analysis did not show any enrichment across the omic views, suggesting that molecules contributing to this factor do not relate to any pathological change, but are rather contributing to physiological activities such as growth.

## Discussion

DMD is a progressive disease for which no cure is available despite heavy investment in the drug development process. A large number of therapeutic compounds ranging from small molecules to gene therapy have been tested in animal, especially in the *mdx* mouse. Despite efforts to harmonize the experimental procedures to assess muscle performance with physiologic and functional tests, the variation in mice performance, differences in study design across laboratories and lack of large confirmatory studies has resulted in a number drugs tested in exploratory preclinical studies being tested in clinical settings [53,54]. The inclusion of objective biomarkers to assess drug efficacy in pre-clinical settings has been proposed as a mechanism to improve the overall success of clinical trials where many failures have been observed [53]. Failures may be the result of under-powered clinical trials or "noisy" outcome measures, however stringent evaluation of drugs at the preclinical stage with objective readouts could contribute to a better selection of medicinal products to be brought forward for clinical development.

The *mdx* mouse is the most widely used model to test therapeutic compounds. Preclinical studies are typically short with exposure to drugs starting early after birth (4–6 weeks of age) and a typical duration of a few weeks. Although some studies in aged *mdx* mice have been reported [55], they are much less represented in the literature. We therefore sought to determine what biological entities describe the dystrophic phenotype in *mdx* mice in the range of 6 to 30 weeks by studying the multiomic profile of mice over this time period with longitudinal blood samples. We completed the sample set by including an additional 2 independent mouse models that have a different gene dosage of the utrophin gene, a dystrophic paralog known to

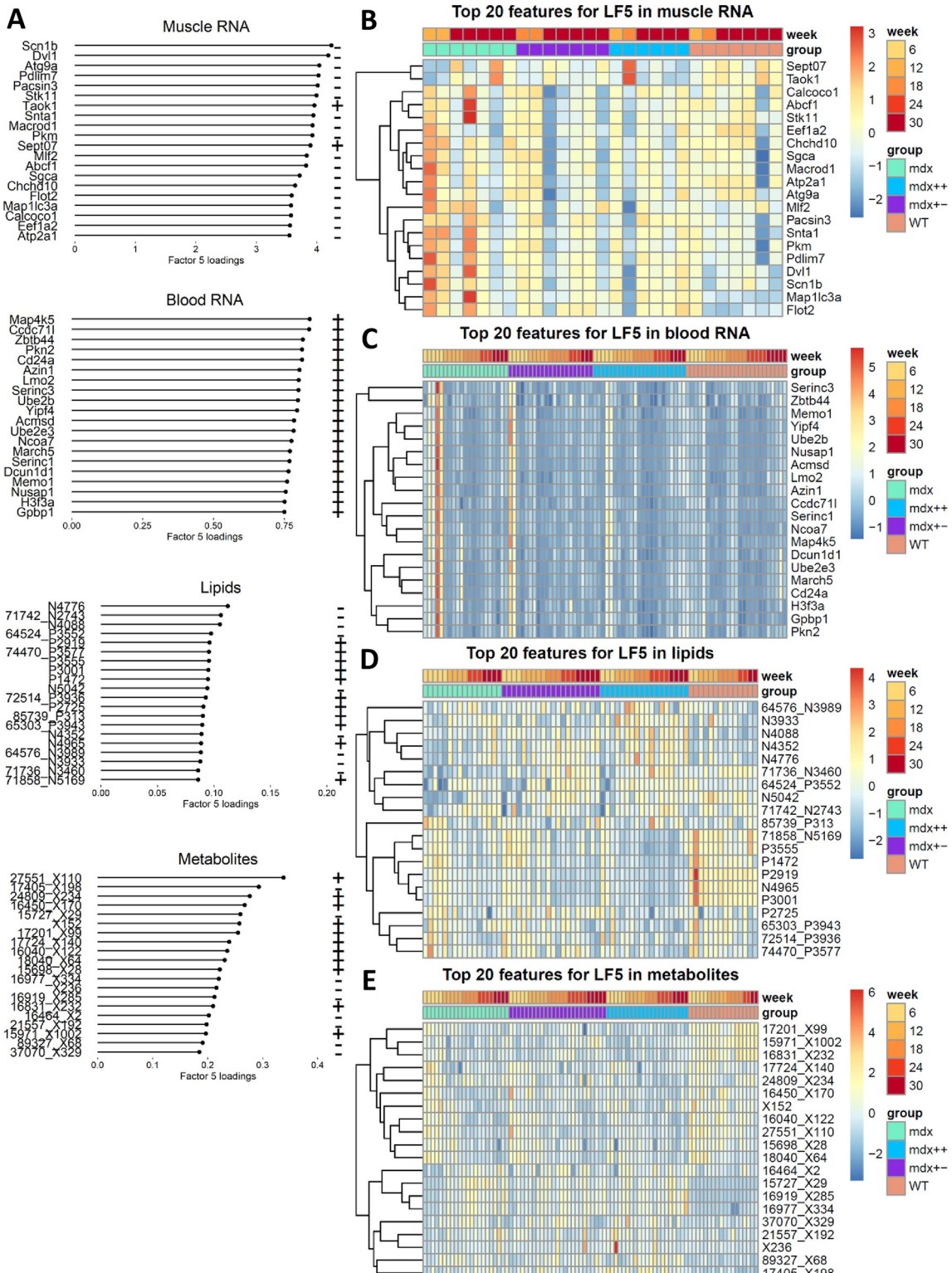

**Fig 5. MOFA factor 3: Top 10 molecules by factor weight in each omic view.** A) Factor 3 weights for the top 10 genes, lipids and metabolites by factor weight in muscle RNA, blood RNA, lipids and metabolites. B-E) Heatmaps showing the distribution of the top 10 molecules by factor weight in muscle RNA (panel B), blood RNA (panel C), lipids (panel D) and metabolites (panel E) across samples.

partly compensate the dystrophic phenotype in mice. Omics such RNA-seq in the affected muscle and blood were used to capture changes that are close to the genetic defect, while lipidomic and metabolomic data ensured to connect the genetics to the phenotypic changes.

Analysis of the multiomic data into the latent space inferred by MOFA allowed to identify LF1 as best discriminator between dystrophic and healthy mice. The muscle RNA-seq signature here was the strongest component highlighting the fibrotic effects following lack of dystrophin; this omic view also highlighted dysregulated lipid metabolism. While the muscle data were only available when mice were sacrificed at 30 weeks of age, blood RNA-seq, lipidomic and metabolomic data were available early on in the experiment, confirming the strong signature apparent in blood as early as in 6 weeks old mice. The dystrophic signature is largely captured by LF1 starting with the affected *Dmd* gene loading high in LF1 along with several pro-fibrotic genes (*Col1a1*, *Col1a2*, *Col3a1*, *Fn1* and *Spp1* to mention a few), regeneration markers (*Myl4*, *Mybph*, *Sparc*, *Igfpb7* and *Gpx1*) along with genes involved in inflammatory and immune response (*Lyz2*, *Spp1*, *Ctsb*). A few genes involved in muscle regeneration such as *Sparc* [56], protein synthesis such as *Igfbp4* [57] and fibrosis such as *Col1a2* [58] showed a direct relationship in muscle and blood gene expression, enabling to connect blood expression data to observations in muscle [58]. A number of genes with high loading in LF1 have been previously reported both as causative and in association with other forms of neuromuscular disorders. It would be interesting to compare the multi-omic profiles to identify disease specific as well as common patterns across different neuromuscular diseases. The lipid signature visible by pathway analysis of the muscle RNA-seq data was mirrored in the lipidomic view, which was the second largest contributor to LF1. Several glycerolipids and glycerophospholipids contributed to the signature along with other lipids such as the reduction of a few sphingomyelins, which is in line with the reduced gene expression levels of the sphingomyelin synthase (also in LF1) in muscle as shown in the muscle RNA-seq. The lipidomic signature peaked at the 6 weeks time point, marking the phase of intense muscle degeneration and regeneration and remained elevated for the whole studied period. Assessing glycerolipids in blood could therefore be a way to obtain information over the dystrophic phenotype in muscle- also in view of the substitution of muscle with adipose tissue observed in DMD patients but absent in mice unless lipid metabolism is further affected by the genetic ablation of the *ApoE* gene in combination with high fat diet. There are unfortunately only a few reports on lipid species in DMD patients, not allowing to compare the lipid metabolism in dystrophic patients and mice. Finally, metabolites further contribute to LF1 especially with nicotinamide N-oxide which is mostly reduced at the early stage of the disease. The reduction can relate to the anti-proliferating activity of this compound [59], which if depleted could reduce the inhibition on cell proliferation and support intense muscle regeneration observed at this time point. The low levels of nicotinamide N-oxide can also relate to the effects on electron transport as also observed in the muscle RNA-seq dataset, as nicotinamide N-oxide is metabolized starting from nicotinamide (a form of vitamin B3) by Cyp2e1 in the liver [60]. Indeed, nicotinamide is the well-known precursor of NAD+/NADH involved in mitochondrial electron transport chain. Nicotinamide N-oxide can also be related to the lipids signature as it has been connected with obesity in mice [61] and type 2 diabetes [62]. Given that nicotinamide N-oxides are antagonists of the CXCR2 receptor [63] and that CXCR2 ligands are over represented in DMD [64], the apparent reduction in serum could relate to the anti-inflammatory action of nicotinamide N-oxide during the early phase of the disease, which is then stabilized from week 12 onwards.

Two factors showed clear associations with time, namely LF4 and LF5. The association of LF4 with time seem to relate with the physiologic growth ongoing in mice between 6 and 30 weeks, as no pathway stood out and no clear grouping effects were observed in the heatmaps

of each omic view. The time effect in LF5 were present for the 6 weeks time point and more heavily present in dystrophic mice especially in the blood RNA-seq view, where pathways such as autophagy of mitochondrion stood out as they were previously shown to be affected in *mdx* mice [65,66] but also in other forms of muscular dystrophy [67]. The metabolomic view highlighted metabolites such as carnosine, creatine and alanine, which are connected to muscle buffering and muscle energetics along with nucleotide metabolism. In this view it was especially interesting to see how patterns at 6 weeks were similar across all groups including dystrophic and WT animals, marking a shared signature for the early time point which is then markedly different in the follow up time points. The difference in progression is in line with the added variance explained by the model with the interaction for LF5 which explained 10% more variants compared to the models where only the main effects are present.

In our experimental design we included *mdx++* and *mdx+-* mice alongside with *mdx* and WT mice. The inclusion of these mice was decided based on a previous report that showed how *mdx+-* mice performed significantly worse compared to *mdx++* [41]. The inclusion of these two groups was therefore aimed to identify molecular signatures that could be associated with the increased phenotype severity observed in *mdx+-* mice. However, our results do not offer evidence for a more severe signature in *mdx+-* mice compared to *mdx++*, suggesting that either our approach is not sensitive enough to capture such differences, or the differences between these mouse models are not as clear as previously anticipated.

A limitation of the study was the lack of muscle samples for the early time points to directly compare the signature with the other omic views. However, it was not possible to obtain muscle tissues at all time points with a longitudinal study design. Another limitation was the blood volume that we were allowed to collect in mice every 6 weeks without affecting animal wellbeing and without sacrificing the mice; the limited volume did not allow us to perform RNA-seq and mass spectrometry analyses on each collected sample, forcing us to match mice to create individual multiomic profiles.

Analysis of the data gathered in this study allowed us to show that it is possible identify common trends that describe the dystrophic phenotype in mice in different omic views. MOFA allowed us to identify genes, metabolites and lipids that may be used to monitor early on in the disease progression, and serve as objective readouts to drug developers to advance therapeutic strategies for DMD.

## Supporting information

**S1 Fig. Representative images of H&E staining performed on tibialis anterior muscle of wt, mdx, mdx utrn++ and mdx utrn+- mice.** Image acquisition was performed at 5X (top panels) and 20X (bottom panel) magnification.
(PDF)

**S2 Fig. Graphical representation of MOFA's matrix decomposition of the data in each view into the product of a view-specific matrix of factor loadings and a matrix of shared latent factors.**
(PDF)

**S3 Fig.** Percentage of total variance explained versus number of LFs for MOFA models fitted including the top 1000 (left) or 2500 (center) genes by variance in blood and muscle RNA seq, or all expressed genes (11243 in muscle and 10349 in blood; right).
(PDF)

**S4 Fig.** Comparison of the percentage of variance explained in the 4 omic views versus number of LFs for MOFA models fitted including the top 1000 (left) or 2500 (center) genes by

variance in blood and muscle RNA seq, or all expressed genes (11243 in muscle and 10349 in blood; right).
(PDF)

**S5 Fig. Expression levels of *Col6a1*, *Col6a2* and *Col6a3* across mice groups in the muscle RNA-seq view.**
(PDF)

**S6 Fig.** Principal component gene set enrichment analysis of LF1 in blood RNA (panel A), muscle RNA (panel B) and metabolites (panel C).
(PDF)

**S7 Fig.** Heatmap of the log-p values of gene sets by factor obtained from the PCGSEA of the GO BP gene ontology in muscle RNA (panel A) and blood RNA (panel B).
(PDF)

**S8 Fig. Comparison of the factor loadings for LF1 in muscle RNA and serum RNA.**
(PDF)

**S1 File. Factor loadings from the fitted MOFA model.**
(XLSX)

**S2 File. Results of the hypothesis tests on difference in mean of each latent factor across mouse groups and between weeks.** Adjusted p-values are obtained using Benjamini-Hochberg's method.
(XLSX)

**S3 File. List of 82 genes previously reported to be associated with neuromuscular disorder among the muscle genes with factor loadings whose absolute value is above 2.**
(XLSX)

**S4 File. List of 501 genes with factor loadings whose absolute value is above 2 that have not been reported to be associated with muscular phenotypes so far.**
(XLSX)

**S5 File. Lipid data (see the "Data availability" section for the RNA-seq and metabolite data).**
(XLSX)

## Author Contributions

**Conceptualization:** Mirko Signorelli, Pietro Spitali.

**Data curation:** Mirko Signorelli, Pietro Spitali.

**Formal analysis:** Mirko Signorelli, Roula Tsonaka, Pietro Spitali.

**Funding acquisition:** Pietro Spitali.

**Methodology:** Mirko Signorelli, Roula Tsonaka, Pietro Spitali.

**Project administration:** Pietro Spitali.

**Resources:** Annemieke Aartsma-Rus, Pietro Spitali.

**Software:** Mirko Signorelli.

**Supervision:** Roula Tsonaka, Annemieke Aartsma-Rus, Pietro Spitali.

**Visualization:** Mirko Signorelli.

**Writing – original draft:** Mirko Signorelli, Pietro Spitali.

**Writing – review & editing:** Mirko Signorelli, Roula Tsonaka, Annemieke Aartsma-Rus, Pietro Spitali.

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
