## [Decision Letter · Decision Letter 0]

13 Jan 2023

PONE-D-22-31031Multiomic characterization of disease progression in mice lacking dystrophinPLOS ONE

Dear Dr. Signorelli,

Thank you for submitting your manuscript to PLOS ONE. After careful consideration, we feel that it has merit but does not fully meet PLOS ONE’s publication criteria as it currently stands. Therefore, we invite you to submit a revised version of the manuscript that addresses the points raised during the review process. Although the reviewers commented favorably on some aspects of your study manuscript, there were significant criticisms that preclude publication. Major weaknesses include 1) The pathological data of several models with several timing used in this study should be shown. 2) LF1 is strongly associated with mouse group and time. The details of LIF1 should be shown such as regeneration factors and novel or unknown factors for muscle pathology.  Please submit your revised manuscript by Feb 27 2023 11:59PM. If you will need more time than this to complete your revisions, please reply to this message or contact the journal office at plosone@plos.org. Please include the following items when submitting your revised manuscript:A rebuttal letter that responds to each point raised by the academic editor and reviewer(s). You should upload this letter as a separate file labeled 'Response to Reviewers'.A marked-up copy of your manuscript that highlights changes made to the original version. You should upload this as a separate file labeled 'Revised Manuscript with Track Changes'.An unmarked version of your revised paper without tracked changes. You should upload this as a separate file labeled 'Manuscript'.

We look forward to receiving your revised manuscript.

Kind regards,

Shinsuke Yuasa

Academic Editor

PLOS ONE

Journal Requirements:

3. We note that Figure 1A  in your submission contain copyrighted images. All PLOS content is published under the Creative Commons Attribution License (CC BY 4.0), which means that the manuscript, images, and Supporting Information files will be freely available online, and any third party is permitted to access, download, copy, distribute, and use these materials in any way, even commercially, with proper attribution. For more information, see our copyright guidelines: http://journals.plos.org/plosone/s/licenses-and-copyright.

a. You may seek permission from the original copyright holder of Figure 1A to publish the content specifically under the CC BY 4.0 license. 

Additional Editor Comments:

Although the reviewers commented favorably on some aspects of your study manuscript, there were significant criticisms that preclude publication. Major weaknesses include 1) The pathological data of several models with several timing used in this study should be shown. 2) LF1 is strongly associated with mouse group and time. The details of LIF1 should be shown such as regeneration factors and novel or unknown factors for muscle pathology.

Reviewers' comments:

Reviewer's Responses to Questions

**Comments to the Author**

1. Is the manuscript technically sound, and do the data support the conclusions?

Reviewer #1: Yes

Reviewer #2: Yes

Reviewer #3: Yes

Reviewer #4: Yes

2. Has the statistical analysis been performed appropriately and rigorously? 

Reviewer #1: Yes

Reviewer #2: Yes

Reviewer #3: I Don't Know

Reviewer #4: Yes

3. Have the authors made all data underlying the findings in their manuscript fully available?

Reviewer #1: Yes

Reviewer #2: Yes

Reviewer #3: Yes

Reviewer #4: Yes

4. Is the manuscript presented in an intelligible fashion and written in standard English?

Reviewer #1: Yes

Reviewer #2: Yes

Reviewer #3: Yes

Reviewer #4: Yes

5. Review Comments to the Author

Reviewer #1: PONE-D-22-31031: Multiomic characterization of disease progression in mice lacking dystrophin

In this article, they tried to find biomarkers for DMD by Multi-Omics Factor Analysis (MOFA) with skeletal muscle samples and blood samples of DMD model mice. They identified some latent factors (LF) to discriminate dystrophic and healthy mice. MOFA enabled to connect the gene expression in dystrophic muscles, pro-fibrotic and energy metabolism alteration. They clearly showed some factors from MOFA to connect gene expression analysis and metabolic analysis.

Major questions and comments

1. In their studies, authors used dystrophic skeletal muscle samples (TA) for Multi-Omics Factor Analysis (MOFA). Readers will want to know how the skeletal muscle pathology of model mice that they used for analysis are. They should show the skeletal muscle pathology (stained sections of TA with H&E stain etc) of mice that they used in their studies.

2. They have identified some LFs which are able to discriminate dystrophic and healthy mice. They mentioned some molecules and genes related to fibrosis, regeneration and inflammation in LF1. Regarding regeneration factors, they should mention how the expression of typical myogenesis-related factors, myoD, myogenin, pax7 etc were changed and related to their analysis if they have any data.

3. They mentioned that there were Col6a1, Col6a2, Col6a3 genes in LF1 in their result.

Ullrich muscular dystrophy (MD) is known to be a different MD from DMD, which caused by abnormal collagen VI, extracellular matrix. In this studies, they analyzed samples of DMD model mice, lack of dystrophin. They should mention that how their results are related to two different MDs.

Reviewer #2: The submitted manuscript entitled “Multiomic characterization of disease progression in mice lacking dystrophin” by Signorelli et al., reported the results of MOFA for disease and age dependent difference in mdx mice models using muscle and blood RNA-seq, blood lipidomics and metabolomics. It is very interesting study to understand the comprehensive signatures of the pathogenesis in mdx mice. The authors found that disease dependent signature as LF1 and age dependent signatures as LF4 and LF5. Especially in LF1, various known genes and molecules were included. It is noteworthy to mention that some gene expression of enzymes as well as their catabolites were simultaneously included. Although this reviewer can agree this manuscript for further process, there are some concerns as below.

1) While this study is important to understand the pathophysiological features of mdx mice, there is no information about human muscular dystrophy. The authors should analyze LF1, 4 or 5 with registered human datasets.

2) In addition, this strategy seems very powerful to understand the entire pathogenesis of mdx mice model, however, there is no information about other models or diseases. The authors should discuss at this point.

3) Especially in LF1, there are well-known genes or molecules, which are related to skeletal muscle pathology. How about novel or unknown gene(s) for muscle pathology among LF1? If there is such genes, the authors must discuss about the relationship between genes and muscle. If not, the authors must describe at this point.

Reviewer #3: This paper attempts to identify factors that correlate with the progression of DMD with age in mdx mice and in mdx+/- and mdx+/+ mice by means of a multi-omics analysis. There were no major problems with the strategy used in this study, and the experiments were conducted logically. However, there are some points that remain unclear, which should be clarified by the authors.

1. It is not clear what is the purpose of using mdx+/- and mdx+/+ in addition to mdx in the first place. In addition, the results and discussion do not seem to clearly mention the benefits obtained by adding these two mice.

2. Conclusions are unclear. It is necessary to clearly state which of the factors involved in the pathological progression of DMD identified in this study could be used as useful markers in the future.

Reviewer #4: In this study, the authors conducted a multiomic analysis of longitudinal trajectories by tissue in DMD mice and identified biomarkers that could observe the etiology and progression of DMD through MOFA analysis. In particular, by identifying pro-fibrotic and energy metabolism alterations, inflammation and lipid signatures that can reveal muscle conditions in the blood, it presents meaningful results in the diagnosis and treatment of DMD.

This study was conducted in a timely manner to understand the pathogenesis, diagnosis and treatment of DMD. In addition, by analyzing multiomic readouts obtained from longitudinal/across tissues in a genius way, it contributed greatly to understanding the cause and progression of disease through a multiomic approach.

This study is a great inspiration for DMD and muscle disease researchers, and is worthy enough to be published with the following minor modifications.

1. (Line 127) Briefly summarize and describe the method of preparation and acquisition of the metabolite and lipids levels.

2. (Line 133) Consistently describe the variable m as a superscript or subscript in the method.

3. (Line 265) Represent the missing E in Figure 2.

4. For non-expert understanding, please provide a graphical summary of MOFA, model training and selection method applied in this study.

5. Please describe the full name other than the symbol of the gene.

6. Using biochemical methods in mouse tissues, please determine the levels of RNAs, metabolites, and lipids involved in fibrosis, muscle regeneration, and immune responses that represent LF1.

6. PLOS authors have the option to publish the peer review history of their article (what does this mean?). If published, this will include your full peer review and any attached files.

Reviewer #1: No

Reviewer #2: No

Reviewer #3: No

Reviewer #4: **Yes: **Joonghoon Park

---

## [Author Response · Author response to Decision Letter 0]

2 Mar 2023

Our point by point reply to the comments from the reviewers can be found in file "rebuttal.pdf".

---

## [Decision Letter · Decision Letter 1]

20 Mar 2023

Multiomic characterization of disease progression in mice lacking dystrophin

PONE-D-22-31031R1

Dear Dr. Signorelli,

We’re pleased to inform you that your manuscript has been judged scientifically suitable for publication and will be formally accepted for publication once it meets all outstanding technical requirements.

Kind regards,

Shinsuke Yuasa

Academic Editor

PLOS ONE

Additional Editor Comments (optional):

Reviewers' comments:

Reviewer's Responses to Questions

**Comments to the Author**

1. If the authors have adequately addressed your comments raised in a previous round of review and you feel that this manuscript is now acceptable for publication, you may indicate that here to bypass the “Comments to the Author” section, enter your conflict of interest statement in the “Confidential to Editor” section, and submit your "Accept" recommendation.

Reviewer #1: All comments have been addressed

Reviewer #2: All comments have been addressed

Reviewer #3: All comments have been addressed

Reviewer #4: All comments have been addressed

2. Is the manuscript technically sound, and do the data support the conclusions?

Reviewer #1: Yes

Reviewer #2: Yes

Reviewer #3: Yes

Reviewer #4: Yes

3. Has the statistical analysis been performed appropriately and rigorously? 

Reviewer #1: Yes

Reviewer #2: Yes

Reviewer #3: Yes

Reviewer #4: Yes

4. Have the authors made all data underlying the findings in their manuscript fully available?

Reviewer #1: Yes

Reviewer #2: Yes

Reviewer #3: Yes

Reviewer #4: Yes

5. Is the manuscript presented in an intelligible fashion and written in standard English?

Reviewer #1: Yes

Reviewer #2: Yes

Reviewer #3: Yes

Reviewer #4: Yes

6. Review Comments to the Author

Reviewer #1: (No Response)

Reviewer #2: The authors adequately responded to the concerns from this reviewer. This reviewer agrees this manuscript for publication in Plos One.

Reviewer #3: The authors responded appropriately to the reviewer's comments. Now the reviewer believes this manuscript could be accepted for publication. Thank you.

Reviewer #4: This revision satisfactorily addresses previous review comments and is a significant improvement. I find this revision to be of sufficient worth to be published.

7. PLOS authors have the option to publish the peer review history of their article (what does this mean?). If published, this will include your full peer review and any attached files.

Reviewer #1: No

Reviewer #2: No

Reviewer #3: No

Reviewer #4: No

---

## [Editor Report · Acceptance letter]

24 Mar 2023

PONE-D-22-31031R1 

Multiomic characterization of disease progression in mice lacking dystrophin 

Dear Dr. Signorelli:

I'm pleased to inform you that your manuscript has been deemed suitable for publication in PLOS ONE. Congratulations! Your manuscript is now with our production department. 

Kind regards, 

on behalf of

Dr. Shinsuke Yuasa 

Academic Editor

PLOS ONE